# *In silico* exploration of potent flavonoids for dengue therapeutics

**Anuraj Phunyal** [1], **Achyut Adhikari**[1]\*, **Jhashanath Adhikari Subin** [2]\*

**1** Central Department of Chemistry, Tribhuvan University, Kirtipur, Kathmandu, Nepal, **2** Bioinformatics and Cheminformatics Division, Scientific Research and Training Nepal Private Limited, Kaushaltar, Bhaktapur, Nepal

☯ These authors contributed equally to this work.
\* achyutraj05@gmail.com (AA); subinadhikari2018@gmail.com (JAS)

**Data Availability Statement:** All relevant data are within the article and its supporting information files.

**Funding:** The author(s) received no specific funding for this work.

## Abstract

Dengue poses a persistent and widespread threat with no effective antiviral drug available till now. Several inhibitors have been developed by targeting the viral non-structural proteins including methyl transferase (NS5) of the dengue virus with possible therapeutic values. In this work, virtual screening, molecular docking, molecular dynamics simulations (200 ns), and assessments of free energy changes have been carried out to identify potential candidates from a database of flavonoids (*ca.* 2000) that may have good curative potential from the disease. The binding affinity of flavonoids, namely Genistein-7-glucoside (FLD1), 6'–O-Acetylgenistin (FLD2), 5,6-dihydroxy-2-(4-hydroxyphenyl)-7-[3,4,5-trihydroxy-6-(hydroxy-methyl)oxane-2-yl]oxychromen-4-one (FLD3), Glucoliquiritigenin (FLD4), and Chrysin-7-O-glucoronide (FLD5) showed the binding affinities of −10.2, −10.2, −10.1, −10.1, −9.9 kcal/mol, respectively, and possessed better values than that of the native ligand (−7.6 kcal/mol) and diclofenac sodium (−7.3 kcal/mol). Drug-likeness of the top five flavonoids were acceptable and no end-point toxicity was hinted by ADMET predictions. The stability of the protein-ligand complexes was accessed from 200 ns molecular dynamics simulations in terms of various geometrical parameters; RMSD, RMSF of residues, Rg, SASA, H-bond, and RPDF. The binding free energy changes of these adducts were calculated by the MM/PBSA solvation model with negative values (from −38.01±7.53 to −17.75±11.03 kcal/mol) indicating the sustained spontaneity of the forward reaction and favorability of the product formation. The geometrical and thermodynamic parameters inferred that the flavonoids could bind at the orthosteric site of the target protein of DENV-2 and could inhibit its functioning, possibly, resulting in the prevention of the disease. Overall, this study highlights the anti-DENV activity of five flavonoids, positioning them as promising candidates for further development as antiviral agents against dengue infection.

## 1. Introduction

The dengue virus (DENV) belongs to the Flaviviridae family and is estimated to impact *ca.* 400 million individuals worldwide annually, primarily through the bite of infectious mosquitoes

**Competing interests:** The authors have declared that no competing interests exist.

[1, 2]. The economic burden of dengue is detrimental, costing USD 8.9 billion yearly [3]. Therefore, prevention and cure of this arboviral disease among others is of dire importance. DENV causes dengue fever, dengue hemorrhagic fever (DHF), dengue shock syndrome (DSS), and also damages various organs [4, 5]. Dengue fever exhibits signs in individuals, including a rise in body temperature reaching 40°C, discomfort in muscles and joints, intense headaches, reddening of the face, appearance of skin rashes, and severe flu-like indications [6]. During DHF, a rapid increase in body temperature is observed, and when entering the critical phase, a notable decline in platelets (below 100,000 cells/μL), white blood cells (WBC), and neutrophil levels are diagnosed [7]. The plasma escape during fever in individuals infected with dengue is a critical condition that can result in DSS [8]. Currently, no specific antiviral medications for treating dengue infection have been developed or approved and it poses a tough challenge in the field of prophylactics design. It can be considered a research gap that mandates prompt filling. Sanofi-Pasteur's tetravalent dengue vaccine (CYD-TDV, Dengvaxia) has been approved for clinical use in several countries [9]. It has raised concerns regarding its overall efficacy in the general population [10]. Therefore, proper, effective, and safe therapeutics for the treatment and cure of dengue are still lacking.

DENV has four serotypes (DENV-1, DENV-2, DENV-3, and DENV-4). Among them DENV-2 is also called severe dengue and is commonly reported in Asia and Latin America [11]. The single-stranded positive-sense RNA genome is its genetic material, *ca*. 11 kb in size [12]. Genomic RNA is translated to a polyprotein which produces three structural proteins, capsid (C), pre-membrane (prM), and envelope (E). These are responsible for forming different parts of the virion structure. Additionally, it encodes seven non-structural proteins (NS1, NS2A/B, NS3, NS4A/B, and NS5) that play a pivotal role in viral RNA replication [13]. This article focuses on NS5 protein of DENV-2 which is considered a good and druggable therapeutic target [14]. It is the largest protein, consisting of *ca*. 900 amino acid residues and has two distinct domains: a methyl transferase domain (MTD) located at the N-terminal and an RNA-dependent RNA polymerase domain (RdRp) at the C-terminal (Fig 1).

The MTD plays a crucial role in preventing viral mRNA degradation from 5'-exoribonucleases of the host. It accomplishes this by adding a methyl group to the mRNA, which protects

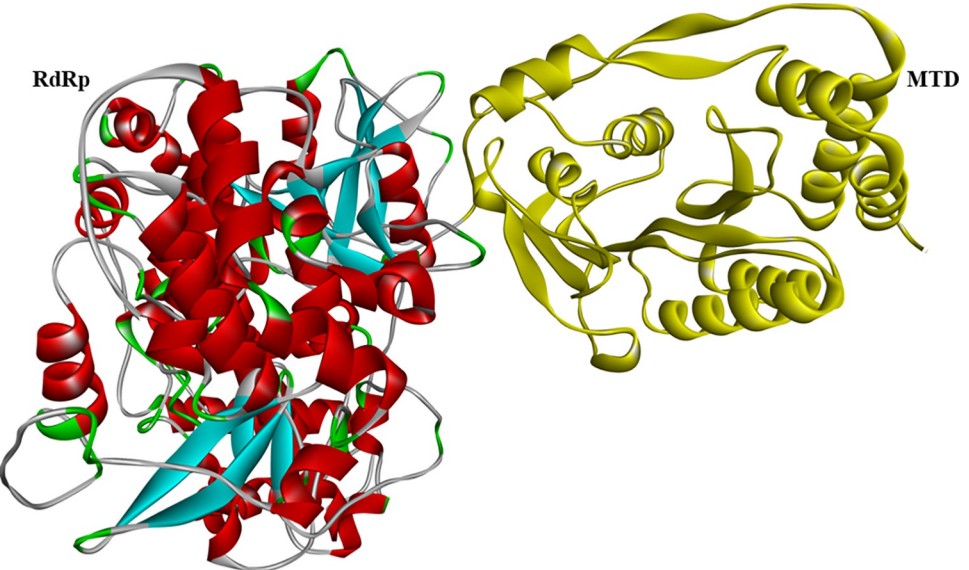

**Fig 1. Cartoon representation of the two domains of NS5 protein of dengue virus chosen as a target structure.**

it from being recognized and degraded by host cellular enzymes. The eukaryotic translation initiation factor can recognize the mRNA required for the continuation of its life cycle [15]. At GTP (guanosine triphosphate) binding site, the methylation occurs at guanosine N-7 position, forming N-7- methylguanosine. At another catalytic site called S-adenosylmethionine (SAM) binding site in MTD, methylation takes place at the ribose, also leading to the formation of 2' -O- methyl-adenosine. SAM (a native ligand) is utilized as the source of methyl groups for this process [16]. It is a secondary metabolite present in virtually all bodily tissues, fluids, and plays a pivotal role in numerous vital functions, including immune system regulation. MTD is a good target and competitive inhibitors of SAM that do not provide methyl groups for protecting mRNA and escaping host immune system are desired for effective therapeutic intervention [17].

Flavonoids, a class of organic molecules, have been reported to possess antiviral activity, making them promising candidates for inhibiting the function of methyl transferase of dengue virus and to cure the disease [18–20]. This work aims to identify NS5 inhibitors from a pool of flavonoids in a low-cost environment. The molecules with the strongest and the most stable binding, with the least or null toxicity, and optimum drug-likeness would be proposed by using different computational methods (molecular docking, molecular dynamics simulation, binding free energy calculations, and ADMET prediction). The results would help in identifying the hit candidates that could be used for further *in vitro* and *in vivo* trials in the drug design and development process.

## 2. Computational experiment

### 2.1. Selection and preparation of ligand database

A dataset of ligands (*ca.* 2000 molecules) was constructed using a similar structure search of the scaffold of a flavonoid and 3D chemical structures were retrieved from the PubChem server in sdf format [21]. The ligands were then prepared for molecular docking by adding polar hydrogen atoms using the Avogadro software v1.1 [22]. After that energy minimization was done with conjugate gradient approach, energy convergence of $10^{-8}$ units, and with UFF force field for 2000 cycles. The geometry optimization was performed multiple times until no significant structural changes were observed ($\Delta E = 0$). The sdf format of ligand was changed to pdb format by using the PyMOL v2.5.4 program and was further converted to pdbqt format followed by the addition of Gasteiger charges [23]. The bond order and the molecular formula were verified for all the compounds.

### 2.2. Protein structure preparation, homology modeling, and validation

The NS5 protein of DENV, with the PDB ID of 5ZQK (resolution of 2.30Å, X-ray diffraction, expression organism: *E. coli*) (https://doi.org/10.2210/pdb5ZQK/pdb) was chosen as the target [24]. The crystal structure was obtained as a pdb file from the Protein Data Bank server (https://www.rcsb.org/). Chain A was chosen and water molecules along with ions and other cofactors were removed. It was followed by the addition of polar hydrogen atoms and Kollman's charge using AutoDockTools v1.5.7 [25]. It was saved in pdbqt format necessary for molecular docking calculations. Since the protein structure was missing some amino acid residues, homology modeling was performed by using the SWISS-MODEL server [26]. The FASTA sequence was downloaded and the BLAST database search method was employed. A template (5zqk.1.A) was generated and structural descriptors like Global Model Quality Estimation (GMQE) along with Quaternary Structure Quality Estimation (QSQE) were obtained [27, 28]. From the selected template, a 3D protein model was generated. It was converted to pdbqt format and subsequently utilized for molecular docking calculations. The SAVES v6.0

web server (https://saves.mbi.ucla.edu/) with three modules namely, ERRAT, VERIFY3D, and PROCHECK (Ramachandran plot) were used to validate the quality of the protein model [29].

### 2.3. Pharmacodynamics and pharmacokinetics

Absorption, distribution, metabolism, excretion (pharmacokinetics), and toxicity (pharmacodynamics) parameters were calculated from the ADMETlab 2.0, ProTox-II, and SwissADME servers, of all the flavonoids and reference drugs (diclofenac sodium) [30–33].

### 2.4. Molecular docking

Molecular docking calculation was performed using the AutoDock Vina v1.1 software, as well as the DockThor and the SwissDock servers for verification of the results [34–36]. In case of first program, AutoDockTools v1.5.7 was used for generating the grid maps of protein and for the visualization. A standard grid box size of 40 Å × 40 Å × 40 Å, the grid center at (57.568, 3.995, and 47.337), an energy range of 4, spacing of 0.375 Å, and the number of modes of 20 with an exhaustiveness of 32 were used. The CASTp server was used to verify the location of the orthosteric site [37]. The top five protein-ligand complexes in terms of binding affinities were saved in pdb format and used for molecular dynamics simulations. Visualization of the binding interaction between protein and ligand was done through Biovia Discovery Studio 2021 software [38].

### 2.5. Molecular dynamics simulation (MDS)

The protein-ligand adduct was simulated by using the GROMACS program, employing the CHARMM27 force field for both the ligand and the receptor [39, 40]. The ligand force field was obtained from the SwissParam server as.zip format [41]. To solvate the system, TIP3P water model was used in a triclinic box (a = 11.504 nm, b = 10.185 nm, c = 11.165 nm, volume = 1308.18 $nm^3$) with a spacing of 12 Å at the sides chosen to minimize the spurious interactions between the periodic images [42]. The system was neutralized with counter ions and an isotonic solution of NaCl (0.15 M) was added. The equilibration was processed in four steps, each of 200 ps, at a physiological temperature of 310 K. The first two steps involved NVT ensemble, while the last two steps involved NPT ensemble. During equilibration, temperature coupling (modified Berendsen thermostat), and pressure coupling (Berendsen) were applied, with PME (particle mesh Ewald) used for long-range Coulomb interactions [43, 44]. A production run was conducted for a duration of 200 ns without imposing any restraints with a step size of 2 fs. Geometrical parameters such as RMSD (Root Mean Square Deviation), RMSF (Root Mean Square Fluctuation), Rg (Radius of Gyration), SASA (Solvent Accessible Surface Area), hydrogen bond count, and RPDF (Radial Pair Distribution Function) were extracted from the MDS trajectory using the inbuilt modules of the GROMACS program.

### 2.6. Binding free energy changes by MM/ PBSA method

Molecular mechanics Poisson-Boltzmann surface area (MM/PBSA) method was used in calculating the changes in binding free energies of the protein-ligand complex according to the following equations [45]

$$\Delta G_{BFE} = \Delta G_{complex} - (\Delta G_{protein} + \Delta G_{ligand}) \tag{1}$$

$$\Delta G_{BFE} = \Delta G_{gas} + \Delta G_{solv} \tag{2}$$

$\Delta G_{gas}$ is the sum of electrostatic ($\Delta E_{EL}$), and van der Waals" energies ($\Delta E_{VDW}$), and the solvation-free energy ($\Delta G_{solv}$) is the sum of polar ($\Delta E_{PB}$), and nonpolar ($\Delta E_{NPOLAR}$) parts [46]

$$\Delta G_{BFE} = \Delta G_{VDW} + \Delta G_{EL} + \Delta G_{PB} + \Delta G_{NPOLAR} \tag{3}$$

Based on the sign of free energy changes, the forward reaction's spontaneity and viability were assessed. Here, 200 frames corresponding to 20 ns of the equilibrated part of the MDS trajectory were adopted.

## 3. Results and discussion

### 3.1. Validation of 3D structure of protein and analysis

Before molecular docking, the model of protein was validated for its structural integrity and robustness from server-based calculations. The overall quality factor from the ERRAT module was found to be 94.22, and from the VERIFY module, the structure of the protein was verified with the parameter of 87.02% (more than 80.00% is considered good). From the PROCHECK module, the Ramachandran plot was obtained and it showed over 90% of the residues in the most favored region (S1 Fig). GMQE and QSQE values were determined to be 0.84 and 0.82 respectively, both exceeding the threshold of 0.7. The active site residues were identified as SER79, GLY109, TRP110, LYS128, ASP154, and VAL155. The results of BLAST obtained from the FASTA sequence of protein have provided additional details of the structure (S2 Fig). The results indicated the structure of the protein to be of high quality and was suitable for computational modeling.

### 3.2. Molecular docking protocol validation

Molecular docking protocol validation was done by docking the native ligand at the active site resulting in heavy-atom RMSD of less than 3 Å relative to the pose in the crystal structure (S3 Fig) [47]. This justified the parameters taken, the algorithm used in obtaining the binding affinities, and the calculated ligand poses.

### 3.3. ADMET analysis

Conducting an ADMET study can help reduce the likelihood of failure or later recall of the drug candidates. The drug-likeness, ADME characteristics, and toxicity profile of the potential candidates were evaluated by a server-based approach. Out of *ca*. 2000 flavonoids taken, only 34 successfully passed the criteria (virtually screened). Many candidates (class 5 toxicity, $LD_{50}$ >2500 mg/kg) showed a lack of hepatotoxicity, carcinogenicity, mutagenicity, immunotoxicity, and cytotoxicity (S1 Table). It was found that the candidates exhibited plasma protein binding (PPB) of less than 90%, indicating suitability and a high therapeutic index (S2 Table). Furthermore, it demonstrated acceptable results (ranging from 0 to 0.3) for skin sensitization and minimal eye corrosion/irritation, almost reaching non-sensitizing levels (S3 Table). The central nervous system permeability (logPS) of the hit candidates were determined to be less than −3, indicating impermeability to the CNS (S4 Table). Additionally, the gastrointestinal (GI) effects of many candidates were also found to be acceptable.

The flavonoids would not inhibit CYP2D6, CYP1A2, CYP2C19, CYP2C9, and CYP3A4 enzymes, which play a vital role in metabolizing xenobiotic and minimizing the negative effects of drug interaction [48]. Also, it does not act as a renal Organic Cation Transporter 2 (OCT2) substrate which is therefore unlikely to have a significant impact on renal clearance. Any prediction of toxicity from *in silico* methods mandates verification by different experimental methods. The reference drug, Diclofenac sodium (class 3, $LD_{50}$ 53 mg/kg)

demonstrated a remarkably elevated PPB rate of 99.21% and thus possessed a relatively low therapeutic index. It is predicted to induce respiratory toxicity and provoke skin sensitization. The high gastrointestinal absorption exhibited by this drug enables effective system distribution and therapeutic effects but it also possesses hepatotoxic properties. The top five flavonoids showed better drug-likeness and lower toxicity compared to both the native ligand and the reference drug. The results usher towards exploration of further steps in the drug design process with the hit candidates.

### 3.4. Molecular docking analysis

34 molecules that passed the ADMET screening, the native ligand and the reference molecule were subjected to molecular docking calculations against NS5 methyl transferase. The possibility of competitive inhibition was explored. The binding affinities ranging from −10.2 to −7.6 kcal/mol, surpassing those of the native ligand (−7.6 kcal/mol) and reference molecule (−7.3 kcal/mol) were obtained, indicating stronger binding (Table 1). FLD1, FLD2, FLD3, FLD4, and FLD5 could be considered as the top 5 candidates based on the docking score (S4 Fig).

The molecular-level details of bonding interaction with the key amino acid residues by different flavonoids along with distances have been extracted (Table 2). Various types of interactions have been found for five different complexes. Notably, FLD1 and FLD2 showed exemplary binding affinities of −10.2 kcal/mol each, and ranked highest among all the molecules tested. FLD1 formed hydrogen bonds with GLY108 (2.35 Å), **GLY109** (2.19 Å), THR127 (2.37 Å), **LYS128** (2.94 Å), GLU134 (2.71 Å), and **ASP154** (2.13 Å), along with GLY81 (3.22 Å), HIS133 (3.80 Å), ARG107 (1.86 Å), **VAL155** (4.97 Å), and ILE170 (3.80 Å) were additional binding site residues involved in other types of associative interactions. Other 9 residues showed van der Waals" interactions in the complex (Fig 2A). Similarly, FLD2 was engaged in hydrogen bonding with GLY81 (2.82 Å), **GLY109** (2.85 Å), **TRP110** (2.39 Å), THR127 (2.29 Å), **LYS128** (2.90 Å), and GLU134 (2.34 Å). Also other non-covalent interactions with GLY81 (3.71 Å), ARG107 (1.55 Å), **LYS128** (4.94 Å), **VAL155** (1.18 Å), HIS133 (3.03 Å), and ILE170 (3.63 Å) were observed. 11 residues with van der Waals' interaction could also be seen in the complex (Fig 2B). The presence of multiple hydrogen bonds in these two complexes could be reason for better docking scores. FLD3 and FLD4 exhibited strong binding affinities of −10.1 kcal/mol each with NS5 protein. FLD3, for instance, formed hydrogen bonds with **SER79** (2.97 Å), ARG107 (2.56 Å), GLY108 (2.28 Å), and GLY171 (2.18 Å). Additional interactions were observed with **LYS128** (4.93 Å), **VAL155** (5.03 Å), and ILE170 (3.61 Å) amino acid residues along with 16 van der Waals' interactions (Fig 2C). The details in the complexes of FLD4 and FLD5 are similar to that of the other three (Fig 3D and 3E). Similar active site residues have been reported in previous studies with the same receptor indicating consistency of the calculations [49, 50]. Despite overall favorable interactions, some unfavorable ones were also present in FLD1, FLD2, FLD4, and FLD5 complexes due to steric hindrance, incompatible geometry, electrostatic repulsion, and solvent effects [51]. Analysis of 3D interactions revealed distinct hydrophobic and hydrophilic regions for the five different adducts (S5 Fig). The brown-colored region indicates hydrophobicity and the blue-colored region denotes hydrophilicity each having nearly compatible type of functional groups of the ligands in their vicinities. The results obtained from two other servers (S5 Table) provide nearly similar inferences as discussed above and henceforth, rationalize the current computational findings. Notably, the top five flavonoids determined from the DockThor server are identical, revealing reproducibility of binding strength from different scoring algorithms. However, the SwissDock server has captured only some top scorers pointing towards the limitations of the use of multiple molecular docking algorithms.

Table 1. List of flavonoids with binding affinities from molecular docking calculations.

| S.N. | Flavonoids (PubChem CID) | Code | Binding Affinity (kcal/mol) |
|------|--------------------------|------|------------------------------|
| 1 | 5284639 | FLD1 | −10.2 |
| 2 | 22288010 | FLD2 | −10.2 |
| 3 | 14332446 | FLD3 | −10.1 |
| 4 | 13473703 | FLD4 | −10.1 |
| 5 | 14135334 | FLD5 | −9.9 |
| 6 | 5213 | FLD6 | −9.8 |
| 7 | 13873666 | FLD7 | −9.8 |
| 8 | 137796321 | FLD8 | −9.8 |
| 9 | 51666248 | FLD9 | −9.8 |
| 10 | 74819354 | FLD10 | −9.7 |
| 11 | 53398699 | FLD11 | −9.7 |
| 12 | 72747690 | FLD12 | −9.5 |
| 13 | 12004622 | FLD13 | −9.5 |
| 14 | 623002 | FLD14 | −9.5 |
| 15 | 73192461 | FLD15 | −9.5 |
| 16 | 5358913 | FLD16 | −9.5 |
| 17 | 5489114 | FLD17 | −9.5 |
| 18 | 4183640 | FLD18 | −9.4 |
| 19 | 25202038 | FLD19 | −9.3 |
| 20 | 51136398 | FLD20 | −9.3 |
| 21 | 3321055 | FLD21 | −9.2 |
| 22 | 3733033 | FLD22 | −9.2 |
| 23 | 5317025 | FLD23 | −9.2 |
| 24 | 73829903 | FLD24 | −9.0 |
| 25 | 3483754 | FLD25 | −8.9 |
| 26 | 74977902 | FLD26 | −8.9 |
| 27 | 4789 | FLD27 | −8.8 |
| 28 | 42607667 | FLD28 | −8.8 |
| 29 | 6169038 | FLD29 | −8.7 |
| 30 | 131752198 | FLD30 | −8.7 |
| 31 | 5080434 | FLD31 | −8.6 |
| 32 | 20979874 | FLD32 | −8.6 |
| 33 | 102421333 | FLD33 | −7.9 |
| 34 | 73009474 | FLD34 | −7.6 |
| 35 | 440426607 | Native ligand (SAM) | −7.6 |
| 36 | 5018304 | Diclofenac sodium | −7.3 |

## 3.5. Molecular dynamics simulation (MDS)

**3.5.1 Root mean square deviation (RMSD).** RMSD profiling of the ligands and that of the backbone were used to analyze the dynamic behavior and stability of the protein-ligand complexes. RMSD of the ligands and those of protein backbones both relative to the protein backbone (least square fitting) of different complexes were extracted from the MDS trajectories (Fig 3A and 3B). The protein backbone exhibited steady RMSD profiles of *ca*. 3 Å for all the cases except for the protein- FLD2 complex. This indicated that the receptor geometry remained stable upon ligand binding and hinted at the druggable nature of the protein. The RMSD profiles of FLD1, FLD3, FLD4, and FLD5 in their complexes showed smooth

**Table 2. Different types of interaction and key amino acid residues of protein (PDB ID 5ZQK) in different complexes.**

| Compounds (PubChem CID) | Interactions | Key amino acid residues | Distance (Å) |
|---|---|---|---|
| **5284639 (FLD1)** | H-bonding | GLY108, **GLY109**, THR127, **LYS128**, GLU134, **ASP154**, | 2.35, 2.19, 2.37, 2.94, 2.71, 2.13, |
| | Alkyl, Pi, ions, Unfavorable | GLY81, HIS133, ARG107, **VAL155**, ILE170 | 3.22, 3.80, 1.86, 4.97, 3.80 |
| | van der Waals' | VAL78, LYS84, GLY104, CYS105, GLY106, **TRP110**, HIS133, VAL153, PHE156 | - |
| **22288010 (FLD2)** | H-bonding | GLY81, **GLY109**,**TRP110**, THR127, **LYS128**, GLU134 | 2.82, 2.85, 2.39, 2.29, 2.90, 2.34 |
| | Alkyl, Pi, ions, unfavorable | GLY81, ARG107, **LYS128** **VAL155**, HIS133, ILE170, | 3.71, 1.55, 4.94, 1.18, 3.03, 3.63 |
| | van der Waals' | **SER79**, ARG80, GLY104, CYS105, GLY106, HIS133, VAL153, **ASP154**, PHE156, ASP169, GLY171 | - |
| **14332446 (FLD3)** | H-bonding | **SER79**, ARG107, GLY108 GLY171 | 2.97, 2.56, 2.28, 2.18 |
| | Alkyl, Pi, ions, unfavorable | **LYS128**, **VAL155**, ILE170 | 4.93, 5.03, 3.61 |
| | van der Waals' | VAL78, GLY81, GLY104, CYS105, GLY106, GLY108, **GLY109**, **TRP110**, SER111 GLU134, VAL153, PHE154, PHE156, ASP169, GLU172 LYS204 | - |
| **13473703 (FLD4)** | H-bonding | CYS105, ARG107, **GLY109** | 2.24, 2.59, 2.76 |
| | Alkyl, Pi, ions, unfavorable | **SER79**, **LYS128**, **VAL155**, **TRP110**, ILE170 | 3.73, 5.06, 3.29, 1.12, 3.63 |
| | van der Waals' | VAL78, GLY81, GLY104, GLY106, GLY108, SER111, THR127, HIS133, GLU134, **ASP154**, PHE156, ASP169 GLU172 | - |
| **14135334 (FLD5)** | H-bonding | CYS105, GLY108, **GLY109**, **TRP110**, **ASP154**, GLY171 LYS204 | 2.51, 2.28, 2.09, 1.87, 2.67, 2.91, 3.08 |
| | Alkyl, Pi, ions, unfavorable | **LYS128**, **VAL155**, ILE170, ARG107, GLY108 | 5.02, 5.21, 3.91, 2.53, 2.59 |
| | van der Waals' | VAL78, **SER79**, GLY81, GLU106, GLY108, SER111, THR127, GLU134, VAL153, **ASP154**, PHE156 | - |

trajectories with values between 2 Å to 5 Å. After *ca.* 140 ns the system attained equilibrium and maintained it until the end indicating greater and continual stability. The RMSD of the FLD2 ligand relative to the protein backbone varied during the production run and could not be equilibrated. The unstable nature of this complex relative to the other four could be inferred. The four ligands could be considered to be localized in the SAM binding pocket of the protein during MDS and similar outcomes have been reported in the previous study [52]. Different snapshots taken at different moments illustrate how a molecular-level understanding may be attained in terms of geometry and dynamics of different systems (S6 Table).

**3.5.2 Snapshots of ligand at the active site of protein during MDS.** Snapshots were retrieved during different instantaneous times of MDS to analyze the orientation (rotational motion) and position (translational motion) of the docked ligands (S6 Table). The images at 0, 50, 100, 150, and 200 ns, revealed that most of the ligands remained at the same location but with different orientations with few exceptions. The geometrical behavior at the molecular-level can be analyzed to justify the nature of the RMSD curve of ligand relative to the backbone.

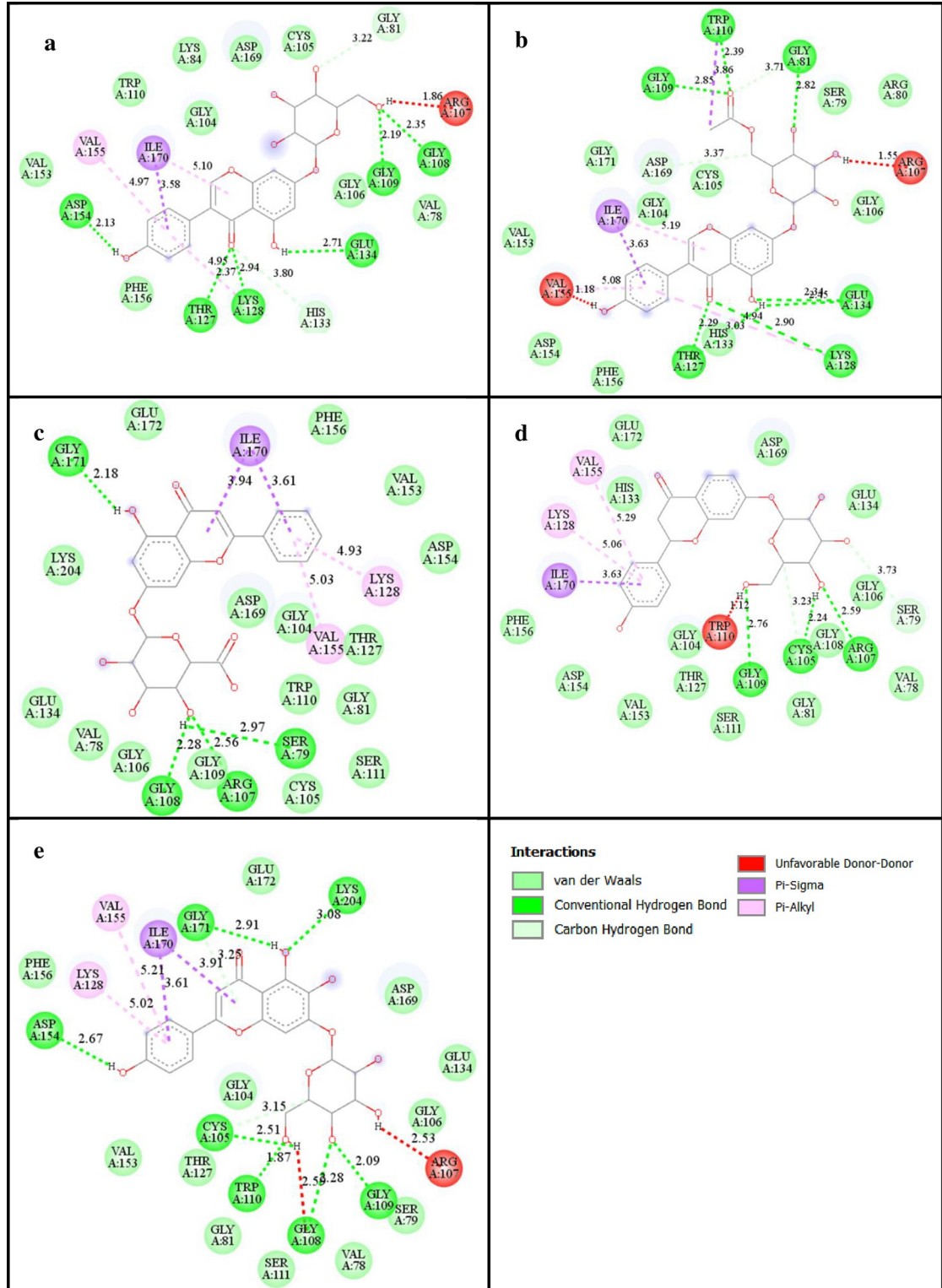

**Fig 2. 2D projection of interaction of ligands with protein.** (a) FLD1 (b) FLD2 (c) FLD3 (d) FLD4, and (e) FLD5.

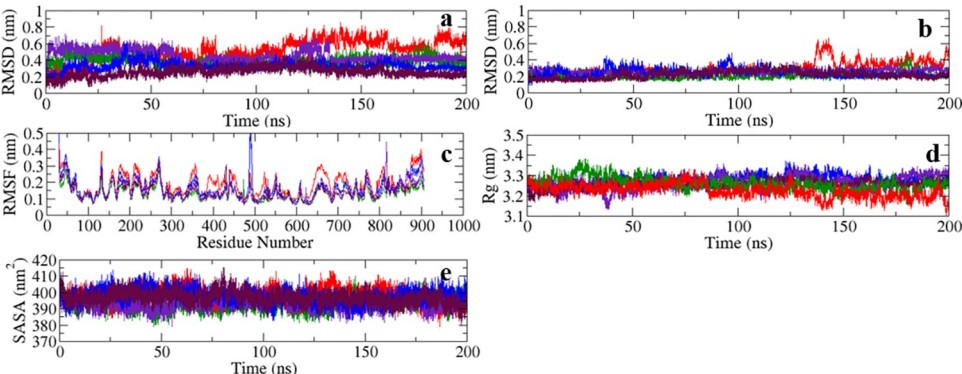

**Fig 3.** RMSD of (a) ligands relative to protein backbone and (b) protein backbones relative to protein backbone, (c) RMSF curves of α-carbon atoms of protein, (d) Rg curves of protein, and (e) SASA of protein in different complexes obtained from 200 ns MDS trajectories Adducts with flavonoids; Indigo curve = FLD1, red curve = FLD2, blue curve = FLD3, maroon curve = FLD4, green curve = FLD5.

In the case of FLD1, the ligand shifted farther apart from the initial position at 50 ns but retained its position at the active site at 100 ns with a change in orientation. It remained at the same location until 200 ns and this behavior is reflected by the nearly smooth RMSD curve after 100 ns. In FLD2, the ligand changed its position and orientation significantly at the active site at 100 and 150 ns, thus causing multiple spikes with irregularities in its RMSD profile. The protein backbone in the protein-FLD2 complex seemed to undergo a significant conformational shift (the alpha-helix at the right part of the image) and was seen as large spikes in the protein backbone RMSD curve. The results indicated that the protein-FLD2 system could not be stabilized at physiological conditions. The images at different times for FLD3, FLD4, and FLD5 complexes denoted minimal delocalization of the ligands with notable rotational motions. Due to the lack of drifting or translational motion from its initial position, the RMSD curves of these ligands were obtained to be nearly smooth in nature. Also, negligible dynamics of the protein backbone were revealed by the snapshots at different instances. The output indicated that the complexes of FLD3, FLD4, and FLD5 were sturdy in nature without undergoing significant disruption. The periodic monitoring of the dynamical behavior of the system provided a clear understanding of the evolution of various molecules that could be correlated to a structural descriptor.

**3.5.3 Root mean square fluctuation (RMSF).** The RMSF was calculated from the MDS trajectory for the α-carbon atoms of the protein, and was found to be below 5 Å in most of the complexes indicating the stability of protein geometry (Fig 3C) [53]. The RMSF of *ca.* 4 Å indicated minimal fluctuation in most amino acid residues of the complexes. However, in the protein-FLD3 complex, residues numbered from 475 to 500 exhibited higher fluctuations at *ca.* 5 Å. This was because of the lack of α-helix or β-sheet secondary structures (constrained geometry with reduced degrees of freedom) and the presence of unbound coils/ loops (unrestrained geometry with higher degrees of freedom) [54]. These regions have no significant interactions with the ligand and large RMSF does not indicate instability. The small fluctuations of the active site residue region at 79 to 155 showed the moderate stability of the receptor upon ligand binding and pointed to negligible effect on the destabilization of the complex. Various RMSF curves conferred that the fluctuations of α-carbon atoms would not have a major impact on the ligand's binding at the active pocket. The adduct's stability would remain unaltered, which would likely result in an inhibition of regular protein functioning.

**3.5.4 Radius of gyration (Rg).** The Rg of protein in various complexes were extracted from the trajectory and it helped to determine the conformational changes of protein structure

(Fig 3D). It gives the average distance from the central axis of the macromolecule to all distributed constituents [55]. The Rg ranged from *ca.* 3.17 to 3.30 nm, hinting at the absence of significant expansion or contraction during the production run except in case of FLD2. The Rg curve of protein in the protein-FLD2 complex showed larger changes after 80 ns until the end, inferring to unstable nature of adduct. The results of the other four adducts imply that there was no considerable receptor expansion or shrinkage in response to ligand binding over the simulated period. The hit flavonoids tended to impart structural intactness and possibly aid in the inhibition of the target protein.

**3.5.5 Solvent accessible surface area (SASA).**   The SASA of protein helps to determine the total wettable area. It was extracted for different structures from the MDS trajectories and was found to lie in the range of 385 to 405 nm$^2$ (Fig 3E). The minimal variation seen in the plot due to the lack of notable changes in the surface geometry provided spatial stability to the complexes upon ligand binding. The protein's hydrophobic portion and its shape remained unchanged which led to the consistency in solvent reachable area [56]. This geometrical descriptor augmented the earlier inference of adduct stability for most of the cases.

**3.5.6 Hydrogen bond count.**   The variation of hydrogen bond count formed between the ligands and protein during the MDS plays a pivotal role in assessing the stability of complexes [57]. The larger the number of hydrogen bonds (non-covalent interactions), the greater is the stability of the complex. The highest hydrogen bond count of up to 8 several times during the MDS was observed in the protein-FLD1 complex (Fig 4A). Similarly, the protein-FLD2 complex showed a count of up to 6 hydrogen bonds but with notable variation (Fig 4B). The complexes with FLD3, FLD4, and FLD5 displayed 4 or more hydrogen bonds (Fig 4C–4E). In the case of protein with FLD4, a consistent curve in the hydrogen bond count was observed, indicating its relationship with ligand RMSD (smooth RMSD curve, maroon) (Fig 3). The RMSD of FLD1, FLD3, FLD4, and FLD5 ligands remained below *ca.* 4 Å due to restricted translational and rotational motion because of near consistency in hydrogen bond count. However, the RMSD of FLD2 was above 4 Å and may be attributed to variation in hydrogen bond count with time. The results indicated that both the number of hydrogen bond counts and its maintenance along with other controlling factors played a key role in maintaining the stability of the protein-ligand systems.

**3.5.7 Radial pair distribution function (RPDF).**   The modulation of distance between two entities over time is provided by RPDF. The plots for various protein-ligand complexes have been extracted from the MDS trajectories. The occurrence of single sharp peak (maxima at *ca.* 2.75 nm) for FLD1 complexes indicated localization of center of mass of ligand relative to center of mass of the protein (Fig 5A). The presence of two peaks in case of protein-FLD2 complex clearly showed the occupancy at two locations (Fig 5B). Similar results have been reported in the literature [58]. In case of FLD3 (*ca.* 2.76 nm), FLD4 (*ca.* 2.76 nm), and FLD5 (*ca.* 2.74 nm) ligands, a single and a slightly broader peak point towards localization but to a lesser degree (Fig 5C–5E). Therefore, RPDF (fate of ligand with time relative to protein) was able to determine the stability of different adducts through different mathematical formulations and could eventually support the inferences derived earlier by multiple parameters.

## 3.6. Thermodynamic spontaneity and stability

The MM/PBSA calculation utilizing the generalized Born model (GBn) was employed to determine the binding free energy differences ($\Delta G_{BFE}$) for the equilibrated portion of the trajectory (20 ns, 200 frames) (Table 3) for different adducts [59].

The negative values of binding free energy changes (ΔGBFE) indicated feasibility of the adduct formation in all the simulated complexes. The lowest binding free energy change (kcal/

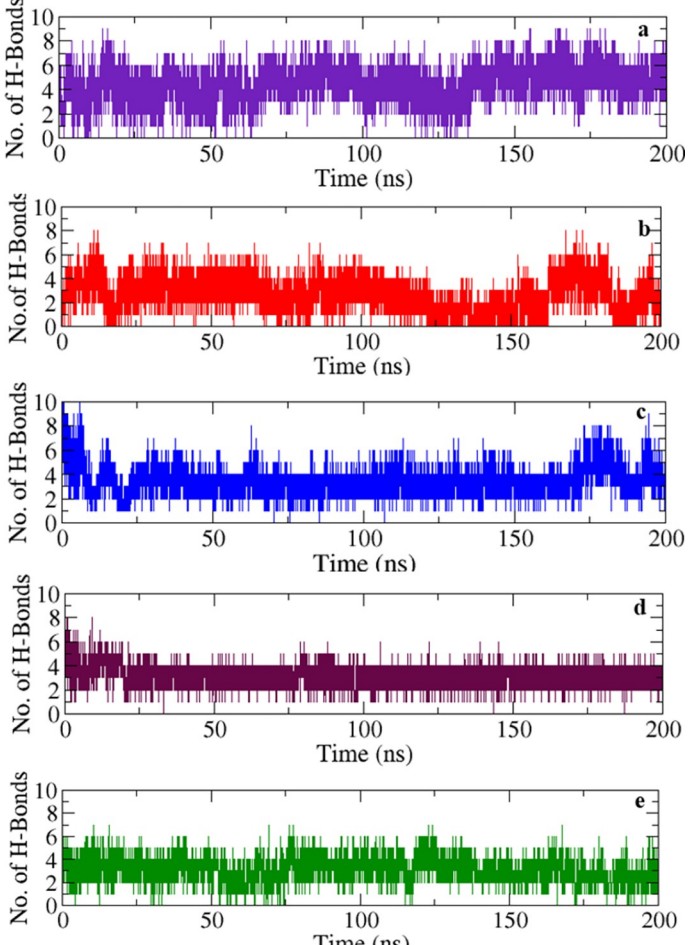

**Fig 4. Number of hydrogen bonds between protein and ligands in different complexes monitored during 200 ns production runs.** Adducts with flavonoids; Indigo line = FLD1, red line = FLD2, blue line = FLD3, maroon line = FLD4, green line = FLD5.

mol) was observed with the protein-FLD3 complex (−38.01±7.53), followed by the complexes of FLD1 (−34.38±5.32), FLD4 (−32.91±5.32), FLD5 (−22.47±6.12), and FLD2 (−17.75±11.03). The frame–by–frame, $\Delta G_{BFE}$ of 20 ns of the equilibrated part of the trajectory is depicted (S6 Fig). The moving average was always negative for most of the complexes implying sustained thermodynamic spontaneity of the reactions in the forward direction. An exception was seen for the protein-FLD2 complex for a very small period. The spontaneous nature of the complex formation reaction indicated that the ligand and the protein preferred to form a complex instead of remaining unattached. Consequently, the interactions between the receptor (PDB ID: 5ZQK), and the flavonoids (FLD1, FLD3, FLD4, and FLD5) were deemed favorable, ensuring the stability of the protein-ligand complexes throughout the production run.

The hit candidates formed spatially and thermodynamically stable protein-ligand complexes as determined from various descriptors extracted from MDS trajectories (cumulative 1 μs). These could notably inhibit or modulate the functioning of methyl transferase by hindering the transfer of methyl group to NS5 protein required for the survival of virus. The preliminary results point towards the requirement of experimental verification for the confirmation of the inferences.

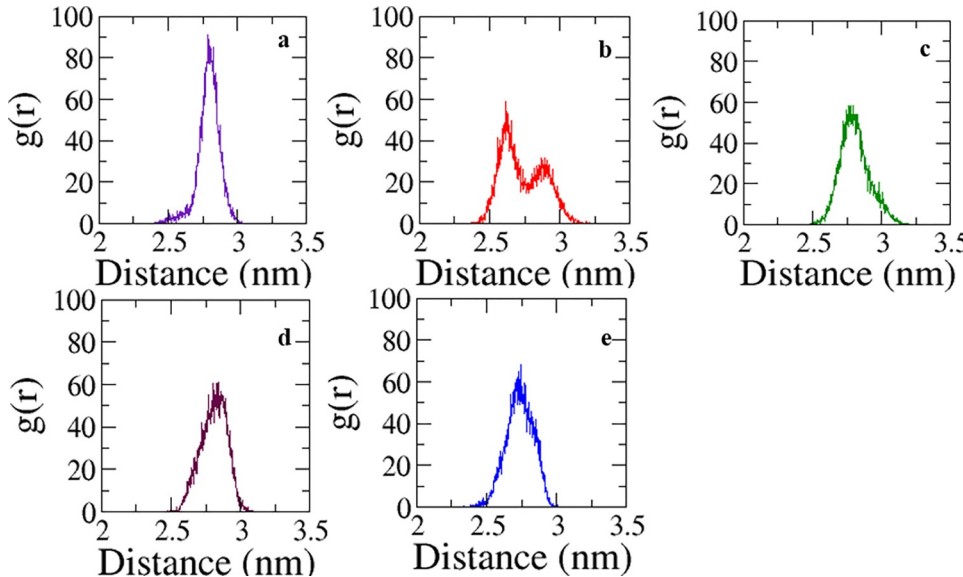

**Fig 5. The radial pair distribution function of the ligand and protein in five different complexes extracted from MDS trajectory.** (a) FLD1, (b) FLD2, (c) FLD3, (d) FLD4, and (e) FLD5; a sharp single peak indicates the localization of the ligand at a definite site in the protein pointing towards the spatial stability of the adduct.

**Table 3. Change in binding free energies (kcal/mol) of complexes with different components.**

| Code | $\Delta E_{VDW}$ | $\Delta E_{EL}$ | $\Delta E_{PB}$ | $\Delta E_{NPOLAR}$ | $\Delta G_{GAS}$ | $\Delta G_{SOLV}$ | $\Delta G_{BFE}$ |
|------|------|------|------|------|------|------|------|
| FLD1 | −36.07±3.31 | −37.87±12.24 | 43.87±7.54 | −4.3±0.11 | −73.94±10.79 | 39.57±7.54 | −34.38±5.32 |
| FLD2 | −42.37±2.69 | −14.89±16.74 | 44.07±20.72 | −4.56±0.26 | −57.26±16.62 | 39.51v20.52 | −17.75±11.03 |
| FLD3 | −45.67±3.16 | −55.73±15.29 | 67.86±11.88 | −4.47±0.10 | −101.40±15.38 | 63.39±11.84 | −38.01±7.53 |
| FLD4 | −44.95±2.63 | −40.12±5.35 | 56.55±5.62 | −4.39±0.09 | −85.07±5.20 | 52.16±5.60 | −32.91±5.32 |
| FLD5 | −34.49±4.29 | −33.49±14.29 | 50.29±11.10 | −4.77±0.14 | −67.99±12.84 | 45.52±11.06 | −22.47±6.12 |

## 4. Conclusions

The potential of flavonoids to inhibit a promising target, NS5 methyl transferase protein of dengue virus (DENV-2), was investigated by different computational methods. Out of 34 flavonoids screened, 33 exhibited higher binding affinity with NS5 protein compared to both the native ligand and a reference drug, diclofenac sodium (employed to lessen the symptoms only). MDS of the top 5 protein-ligand candidates showed good structural stability. The thermodynamical consideration hinted at the sustained spontaneity of the complex formation reaction and the maintenance of the ligand's pose at the orthosteric site of the receptor. Based on the preliminary results (lowest deviation below 2 Å, smoothest RMSD profile with *ca*. 0.5 Å variation, and comparable free energy changes of −32.91±5.32 kcal/mol), FLD4 could be considered a hit molecule. However, all the top candidates, FLD1, FLD3, FLD4, and FLD5 showed promising potential and are recommended for additional *in vitro* and *in vivo* trials to warrant their inhibition capabilities against dengue protein.

## Supporting information

**S1 Fig. Structure of protein validation from SAVES 6.0 server.**
(TIF)

**S2 Fig. Model-Template alignment of protein.**
(TIF)

**S3 Fig. Superimposition of co-crystallized ligand (yellow) and docked ligand (grey-red) obtained from calculations.** Heavy-atom RMSD = 2.618 Å.
(TIF)

**S4 Fig. 2D chemical structures of top five ligands. Based on binding affinity.**
(TIF)

**S5 Fig. Docking pose of ligands in the cavity with a hydrophobic surface of protein.** (a) FLD1 (b) FLD2 (c) FLD3 (d) FLD4, and (e) FLD5.
(TIF)

**S6 Fig. Variation of binding free energy changes of various adducts of protein with FLD1, (b) FLD2, (c) FLD3, (d) FLD4, and (e) FLD5. The moving average is shown in red.**
(TIF)

**S1 Table. Toxicity of the compounds from ProTox-II.**
(DOCX)

**S2 Table. ADMET properties of the compounds from ADMETlab 2.0.**
(DOCX)

**S3 Table. Toxicity from ADMElab 2.0.**
(DOCX)

**S4 Table. ADMET properties from pkCSM server.**
(DOCX)

**S5 Table. Binding affinity of compounds from servers.**
(DOCX)

**S6 Table. Snapshots of five protein-ligand complexes at different times during MDS.** Orientation and location of ligands relative to protein backbone are to be monitored; ligand is shown in ball and stick model and protein in cartoon representation.
(DOCX)

## Acknowledgments

The authors would like to acknowledge Prof. Dr. Rameshwar Adhikari (Tribhuvan University) for partial support of office space. Also, insight into plots construction by Assoc. Prof. Dr. Tika Ram Lamichhane (Tribhuvan University) is highly appreciated.

## Author Contributions

**Conceptualization:** Achyut Adhikari.

**Data curation:** Anuraj Phunyal.

**Formal analysis:** Anuraj Phunyal, Jhashanath Adhikari Subin.

**Investigation:** Anuraj Phunyal, Jhashanath Adhikari Subin.

**Methodology:** Anuraj Phunyal.

**Resources:** Jhashanath Adhikari Subin.

**Software:** Anuraj Phunyal.

**Supervision:** Achyut Adhikari.

**Validation:** Anuraj Phunyal.

**Visualization:** Anuraj Phunyal, Jhashanath Adhikari Subin.

**Writing – original draft:** Anuraj Phunyal.

**Writing – review & editing:** Anuraj Phunyal, Achyut Adhikari, Jhashanath Adhikari Subin.

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
