## [Decision Letter · Decision Letter 0]

12 Aug 2024

PONE-D-24-11567In silico exploration of potent flavonoids for dengue therapeuticsPLOS ONE

Dear Dr. Phunyal,

Thank you for submitting your manuscript to PLOS ONE. After careful consideration, we feel that it has merit but does not fully meet PLOS ONE’s publication criteria as it currently stands. Therefore, we invite you to submit a revised version of the manuscript that addresses the points raised during the review process.

We look forward to receiving your revised manuscript.

Kind regards,

Ranjan K. Mohapatra, PhD

Academic Editor

PLOS ONE

A clean copy of the edited manuscript (uploaded as the new *manuscript* file)”.

Additional Editor Comments:

I have received the reviewer comments on the manuscript entitled“ In silico exploration of potent flavonoids for dengue therapeutics” submitted by Phunyal and coworkers. The manuscript discusses possible flavonoids for the inhibition of the dengue protein. My suggestions are listed below.

1. Add a detailed discussion on the background of dengue cases. Please refer very recent articles: https://doi.org/10.1002/hsr2.1831; https://doi.org/10.1002/hsr2.2089

2. Make a discussion on why the study on arboviral diseases is necessary in recent days.

3. Very old references are used. Total 78 references. Make upto 50 references. It is a research article.

4. Please see in the introduction---"Currently, there is a lack of specific antiviral medications for treating dengue infection. SanofiPasteur’s tetravalent dengue vaccine (CYD-TDV, Dengvaxia) has been approved for clinical use in several countries [8,9]. It has raised concerns regarding its overall efficacy in the general population [10,11]. Therefore proper, effective, and safe therapeutics for the treatment and cure of dengue are still lacking."----all the references are old. Here written currently???? Refer recent data and reference.

5. As per your findings, which one/two is best, discuss with data in conclusion.

6. Due to lack of specific antiviral medications for treating dengue infection, please discuss other control measures in the conclusion (2/3 lines).

7. Make the references as per journal style.

Please revise your manuscript and submit along with your point by point response.

Thank you.

Sincerely,

AE

Comments from PLOS Editorial Office:

We note that one or more reviewers has recommended that you cite specific previously published works. As always, we recommend that you please review and evaluate the requested works to determine whether they are relevant and should be cited. It is not a requirement to cite these works. We appreciate your attention to this request.

Reviewers' comments:

Reviewer's Responses to Questions

**Comments to the Author**

1. Is the manuscript technically sound, and do the data support the conclusions?

Reviewer #1: Yes

Reviewer #2: Yes

2. Has the statistical analysis been performed appropriately and rigorously? 

Reviewer #1: N/A

Reviewer #2: N/A

3. Have the authors made all data underlying the findings in their manuscript fully available?

Reviewer #1: Yes

Reviewer #2: Yes

4. Is the manuscript presented in an intelligible fashion and written in standard English?

Reviewer #1: Yes

Reviewer #2: Yes

5. Review Comments to the Author

Reviewer #1: 1. A decrease in platelet counts below 100000 cells/micro L, and a decrease in WBC and Neutrophil are diagnostic parameters of dengue fever (Potts JA, Rothman AL. Clinical and laboratory features that distinguish dengue from other febrile illnesses in endemic populations. Trop Med Int Health. 2008 Nov;13(11):1328-40. doi: 10.1111/j.1365-3156.2008.02151.x. Epub 2008 Sep 16. PMID: 18803612; PMCID: PMC2756447.).

It should be included in the Introduction.

2. The Authors may also refer to Mohapatra et al (2023) while discussing the role of flavonoids as antiviral agents along with ref.24. (Mohapatra, P. K., Chopdar, K. S., Dash, G. C., Mohanty, A. K., & Raval, M. K. (2023). In silico screening and covalent binding of phytochemicals of Ocimum sanctum against SARS-CoV-2 (COVID 19) main protease. Journal of Biomolecular Structure and Dynamics, 41(2), 435–444. https://doi.org/10.1080/07391102.2021.2007170)

Authors may also consider referring the following article along with Ref25

Thames et al. Synthesis and biological evaluation of novel flexible nucleoside analogues

that inhibit flavivirus replication in vitro. Bioorganic & Medicinal Chemistry 28 (2020) 115713

3. Discussion of Sinefungin a natural mimic of SAM/SAH is relevant.

Ref Yebra et al. The effect of sinefungin and synthetic analogues on RNA and DNA methyltransferases from Streptomyces. The Journal of Antibiotics 1991; 44 (10): 1141- 1147.

4. Respiratory toxicity of FLD1 (0.817) needs discussion on whether FLD1 to be included in short listed prodrugs.

5. The resolution of Figures appearing in the MS is not acceptable. It needs improvement.

Reviewer #2: The MS entitled“ In silico exploration of potent flavonoids for dengue therapeutics” by Phunyal et al discusses possible flavonoids for the inhibition of the dengue protein. The MS is well composed and written.

Few general comments need to be addressed by the authors.

"The database of ca. 2000 molecules was screened based on toxicity criteria from the ADMETlab

2.0 server "– Why only "2000 "phyto compounds are only screened?

Why homology modelling of the receptor is required? The blast result may be provided in the supplementary section

In material method section. “homology modeling” the template name is missing.

To find the potential candidates for dengue therapeutics molecular docking was performed using

Autodock Vina software (version 1.1) was used for molecular docking [49]

Sentenced need to be reconstructed

Grid size in the Molecular Dynamics section in material methods should be mentioned.

Figures should have better clarity and Fig 09 required be re-adapted with better clarity and the Y- axis in specific.

Authors should look into the spelling and grammatical mistakes.

Reviewer #3:

Figures seem very low resolution without proper description of legends. Four- five figures will be standard.

Please apply three to for docking server and validate results to get best output.

Please clear about control group run. It is mandatory for docking and simulation.

500ns MD simulation is highly appreciated, if it is not possible please add more data analysis based on statistic.

Introduction

Please add few lines for epidemiology :https://doi.org/10.1002/hsr2.1831;

Please add references: The single stranded positive-sense RNA genome is its genetic material, approximately 11 kb in size. DOI: 10.1097/MS9.0000000000000623

Please add research gaps regarding this research.

Method:

For protein structure validation, the SAVES v6.0 web server (https://saves.mbi.ucla.edu/) with three modules namely, ERRAT [37], VERIFY3D [38], and PROCHECK [39] (Ramachandran plot) were used to identify protein sequences and assessed the quality of protein structure. Please add accuracy level in the result part.

Please perform two or more docking servers.

Result

Please add data in supplementary files.

Please follow same format for number and digit writing: The overall quality factor from the ERRAT module of the protein was found to be 94.2197%, and from the VERIFY module, the structure of the protein was verified (87.02%)

Figures

Increase resolution. Suggestion: Add ppt slides or follow journal rules. Please lessen figures and transfer in supply.

6. PLOS authors have the option to publish the peer review history of their article (what does this mean?). If published, this will include your full peer review and any attached files.

Reviewer #1: **Yes: **Mukesh Kumar Raval

Reviewer #2: **Yes: **Pranab Kishor Mohapatra

---

## [Author Response · Author response to Decision Letter 0]

16 Sep 2024

All the comments and suggestions have been thoroughly addressed and has been mentioned in response to reviewers.

---

## [Decision Letter · Decision Letter 1]

4 Oct 2024

PONE-D-24-11567R1In silico exploration of potent flavonoids for dengue therapeuticsPLOS ONE

Dear Dr. Adhikari Subin,

Thank you for submitting your manuscript to PLOS ONE. After careful consideration, we feel that it has merit but does not fully meet PLOS ONE’s publication criteria as it currently stands. Therefore, we invite you to submit a revised version of the manuscript that addresses the points raised during the review process.

We look forward to receiving your revised manuscript.

Kind regards,

Ranjan K. Mohapatra, PhD

Academic Editor

PLOS ONE

Journal Requirements:

Reviewers' comments:

Reviewer's Responses to Questions

**Comments to the Author**

1. If the authors have adequately addressed your comments raised in a previous round of review and you feel that this manuscript is now acceptable for publication, you may indicate that here to bypass the “Comments to the Author” section, enter your conflict of interest statement in the “Confidential to Editor” section, and submit your "Accept" recommendation.

Reviewer #1: All comments have been addressed

Reviewer #2: All comments have been addressed

Reviewer #3: (No Response)

2. Is the manuscript technically sound, and do the data support the conclusions?

Reviewer #1: Yes

Reviewer #2: Yes

Reviewer #3: Yes

3. Has the statistical analysis been performed appropriately and rigorously? 

Reviewer #1: Yes

Reviewer #2: N/A

Reviewer #3: Yes

4. Have the authors made all data underlying the findings in their manuscript fully available?

Reviewer #1: Yes

Reviewer #2: Yes

Reviewer #3: Yes

5. Is the manuscript presented in an intelligible fashion and written in standard English?

Reviewer #1: Yes

Reviewer #2: Yes

Reviewer #3: Yes

6. Review Comments to the Author

Reviewer #1: The revision has improved the quality of the article. Authors have addressed all the queries satisfactorily.

Reviewer #2: The authors addressed the required queries raised in the 1st round review process. The MS may be accepted in the present form.

Reviewer #3: Please decrease the number of figures:

1. Amalgamate fig a1 and 2

2. Delete fig 3 or move in supplymentary file

3. Move fig 5 to supplymentary file

4. Fig 6-10 (add all of them like 5a, 5b, 5c, and so on)

7. PLOS authors have the option to publish the peer review history of their article (what does this mean?). If published, this will include your full peer review and any attached files.

Reviewer #1: **Yes: **Mukesh Kumar Raval

Reviewer #2: **Yes: **Pranab Kishor Mohapatra

Reviewer #3: **Yes: **Md Aminul Islam

---

## [Author Response · Author response to Decision Letter 1]

9 Oct 2024

Response: We have reviewed the reference list for ensuring that it is complete and correct.

Review Comments to the Author

Reviewer #1: The revision has improved the quality of the article. Authors have addressed all the queries satisfactorily.

Response: Thank you for your positive feedback.

Reviewer #2: The authors addressed the required queries raised in the 1st round review process. The MS may be accepted in the present form.

Response: We appreciate your recommendation for accepting the manuscript in its current form.

Reviewer #3: 

Please decrease the number of figures:

1. Amalgamate fig a1 and 2

Response: We have moved Fig 2 to the supplementary file as S3 Fig to decrease the figure count. We could not amalgamate Fig 1 (part of the Introduction) and Fig 2 (part of the Results and Discussion) because these are present at different manuscript sections.

2. Delete fig 3 or move in supplementary file

Response: Fig 3 has been moved to the supplementary file.

3. Move fig 5 to supplementary file

Response: Fig 5 has been transferred to the supplementary file.

4. Fig 6-10 (add all of them like 5a, 5b, 5c, and so on)

Response: Thank you for your suggestion. Figures 6 to 9 (Fig 10 already has a number of figures) have been consolidated and presented as a single figure with subparts (according to journal style). Consequently, the reduction in the number of figures in the main manuscript has been done. Thank you.

---

## [Decision Letter · Decision Letter 2]

24 Oct 2024

In silico exploration of potent flavonoids for dengue therapeutics

PONE-D-24-11567R2

Dear Dr. Adhikari Subin,

We’re pleased to inform you that your manuscript has been judged scientifically suitable for publication and will be formally accepted for publication once it meets all outstanding technical requirements.

Kind regards,

Ranjan K. Mohapatra, PhD

Academic Editor

PLOS ONE

Additional Editor Comments (optional):

*Comments from PLOS Editorial Office: We note that one reviewer has recommended that you cite specific previously published works. As always, we recommend that you please review and evaluate the requested works to determine whether they are relevant and should be cited. However, please note that no further changes are required at this stage for publication of the manuscript.*

Reviewers' comments:

Reviewer's Responses to Questions

**Comments to the Author**

1. If the authors have adequately addressed your comments raised in a previous round of review and you feel that this manuscript is now acceptable for publication, you may indicate that here to bypass the “Comments to the Author” section, enter your conflict of interest statement in the “Confidential to Editor” section, and submit your "Accept" recommendation.

Reviewer #3: All comments have been addressed

2. Is the manuscript technically sound, and do the data support the conclusions?

Reviewer #3: Yes

3. Has the statistical analysis been performed appropriately and rigorously? 

Reviewer #3: Yes

4. Have the authors made all data underlying the findings in their manuscript fully available?

Reviewer #3: Yes

5. Is the manuscript presented in an intelligible fashion and written in standard English?

Reviewer #3: Yes

6. Review Comments to the Author

Reviewer #3: Please check 2.4 portion. Authors may add thermodynamics portion. (https://doi.org/10.3389/fphar.2024.1465827)

7. PLOS authors have the option to publish the peer review history of their article (what does this mean?). If published, this will include your full peer review and any attached files.

Reviewer #3: **Yes: **Md Aminul Islam

---

## [Editor Report · Acceptance letter]

30 Oct 2024

PONE-D-24-11567R2 

PLOS ONE

Dear Dr. Adhikari Subin, 

I'm pleased to inform you that your manuscript has been deemed suitable for publication in PLOS ONE. Congratulations! Your manuscript is now being handed over to our production team.

Kind regards, 

on behalf of

Dr. Ranjan K. Mohapatra 

Academic Editor

PLOS ONE